

# The comparison of cold-water immersion and cold air therapy on maximal cycling performance and recovery markers following strength exercises

Kane J. Hayter[1], Kenji Doma[1], Moritz Schumann[2] and Glen B. Deakin[1]

[1] Sport and Exercise Science, James Cook University, Townsville, Queensland, Australia
[2] Department of Biology of Physical Activity, University of Jyväskylä, Jyväskylä, Finland

## ABSTRACT

This study examined the effects of cold-water immersion (CWI) and cold air therapy (CAT) on maximal cycling performance (i.e. anaerobic power) and markers of muscle damage following a strength training session. Twenty endurance-trained but strength-untrained male (n = 10) and female (n = 10) participants were randomised into either: CWI (15 min in 14 °C water to iliac crest) or CAT (15 min in 14 °C air) immediately following strength training (i.e. 3 sets of leg press, leg extensions and leg curls at 6 repetition maximum, respectively). Creatine kinase, muscle soreness and fatigue, isometric knee extensor and flexor torque and cycling anaerobic power were measured prior to, immediately after and at 24 (T24), 48 (T48) and 72 (T72) h post-strength exercises. No significant differences were found between treatments for any of the measured variables (p > 0.05). However, trends suggested recovery was greater in CWI than CAT for cycling anaerobic power at T24 (10% ± 2%, ES = 0.90), T48 (8% ± 2%, ES = 0.64) and T72 (8% ± 7%, ES = 0.76). The findings suggest the combination of hydrostatic pressure and cold temperature may be favourable for recovery from strength training rather than cold temperature alone.

## INTRODUCTION

A growing body of evidence suggests that the application of cold-water immersion (CWI) following strength exercise may accelerate recovery to alleviate symptoms of delayed onset muscle soreness (DOMS) and muscle damage (*Leeder et al., 2012*). The reported benefits include peripheral vasoconstriction (*Karunakara, Lephart & Pincivero, 1999*) which increases metabolite removal (*Cochrane, 2004*) and a decrease in oedema formation (*Dolan et al., 1997*; *Kowal, 1983*).

Of the studies that have examined CWI effects following strength exercises, the comparator groups have typically involved active recovery (*Roberts et al., 2014*), warm water immersion (*Vaile et al., 2008*) and contrast therapy (i.e. alternating between warm and cold water) (*Vaile et al., 2008*). Whilst these conditions demonstrate the influence of temperature on recovery, it does not account for contribution of hydrostatic pressure during water immersion.

Corresponding author
Kenji Doma, kenji.doma@jcu.edu.au

One method of accounting for hydrostatic pressure on recovery could be to compare recovery dynamics between hydrotherapy (i.e. in cold water) to non-hydrotherapy (i.e. in cold air) conditions. Recent studies have reported that cryostimulation (i.e. exposure to cold air) at extremely cold conditions (e.g. under −100 °C) may accelerate recovery and minimise inflammatory responses responsible of inducing DOMS following muscle damaging exercises (*Ziemann et al., 2014*) and mitigate signs of overtraining during periods of intense training (*Schaal et al., 2015*). However, the typical length of exposure for cryostimulation is approximately 2–3 min (*Costello et al., 2015*) due to the extreme conditions, which is substantially less than the typical exposure for CWI of 10–15 min (*Versey, Halson & Dawson, 2013*). Subsequently, comparisons for the effect of recovery between the conventional method of cold air treatment (CAT) and CWI is at present difficult due to large differences in technique. However, by standardizing the temperature between CWI and CAT (e.g. 4 °C in both hydrotherapy and non-hydrotherapy conditions), the length of exposure between treatments could be equated thereby providing the opportunity to determine the contribution of hydrostatic pressure during CWI. To date, comparison between such conditions is limited, particularly following a typical strength training session.

Furthermore, it is unknown whether the benefits of CWI are also reflected in sprint-based exercises (i.e. anaerobic performance) days following a typical strength training session, particularly in strength-untrained and/or detrained individuals where DOMS may elevate to levels that may impair anaerobic performance. Several studies have reported impaired running (*Doma & Deakin, 2013b*; *Doma & Deakin, 2014*; *Doma et al., 2015*; *Twist & Eston, 2005*) and cycling (*Byrne & Eston, 2002*; *Nieman et al., 2014*) performance at maximal intensities for 24–72 h post strength exercises in strength-untrained individuals. These findings have severe implications for the quality of maximal intensity intermittent training sessions when combined with strength training sessions in the one program, also known as concurrent training (*Hickson, 1980*). In fact, *Doma et al. (2015)* recently showed that the combination of alternating-day strength training and consecutive-day maximal intensity endurance training impaired running performance at maximal effort over the course of a typical micro-cycle of concurrent training, which may possibly be detrimental to optimal long-term adaptations. Subsequently, incorporating recovery modalities, such as CWI, may alleviate acute carry-over effects of fatigue in-between each mode of training session and thereby optimise the quality of high intensity intermittent training sessions. *Rowsell et al. (2014)* previously reported recovery of maximal cycling time-trial and cycling interval 9 h following run training via use of CWI, suggesting that CWI may in fact accelerate recovery and minimise carry-over effects of fatigue on maximal intensity intermittent training sessions several hours post strength training. However, little is known whether benefits of CWI for maximal sprint-based performance are present following typical strength exercises (e.g. leg press, leg extension and leg curls), particularly during periods when DOMS peak (i.e. several days post).

The purpose of the current study was to compare the effect of CWI and cold air treatment (CAT) on maximal cycling performance and post-exercise markers of muscle

damage following a typical strength training session in endurance-trained but strength-untrained individuals.

## METHODS

### Research design

The study was conducted over a 2 week period (Fig. 1) with subjects attending a familiarisation session followed by 4 testing sessions. The familiarisation session occurred in the first week and allowed participants to become familiar with the testing procedures and equipment as well as completing a 6 Repetition Maximum (6RM) assessment. The 6RM assessments followed previously described guidelines (*Baechle & Earle, 2008*) for incline leg press (Maxim, MPL 701, Adelaide, Australia), leg extensions and leg curls (Maxim, P 5021, Adelaide, Australia). After a minimum of 4 days of rest, the participants returned to the laboratory and completed a strength training session. Indirect markers of muscle damage, muscle force generation capacity and maximal cycling performance were measured prior to (T0) and immediately post (T1) as well as 24 (T24), 48 (T48) and 72 (T72) h post the strength training session. Immediately following the T1 testing time point, participants undertook the recovery protocol either as an intervention by submerging into water (i.e., CWI) or as a CAT. During each visit, participants underwent a standardised warm-up consisting of five minutes of stationary cycling (Ergomedic 828E; Monark, Vansbro, Sweden). Biological variations were controlled by conducting the strength training sessions at the same time of day, refraining from caffeine or food intake at least 2 h prior to testing and high intensity physical activity for a minimum of 24 h prior.

### Subjects

Twenty strength-untrained but moderately endurance trained males (n = 10) and females (n = 10) volunteered to participate in this study. The participants were recreational endurance trained males and females (e.g. runners, cyclists) who had been participating in moderate-high intensity endurance exercise at least twice a week for the previous 12 months and had not performed lower body strength trainings sessions for at least 6 months. The participants were manually matched paired by gender, age and muscular strength and then allocated at random to either a cold water immersion (CWI; age 25.3 ± 6.0 years, height 170.9 ± 8.1 cm, body mass 70.2 ± 8.9 kg, leg press strength 169.5 ± 71.7 kg) or CAT (age 22.5 ± 3.9 years, height 171.6 ± 10.5 cm, body mass 70.4 ± 13.9 kg, leg press strength 167 ± 63.3 kg) group and were matched by gender, age and muscular strength. Whilst previous CWI studies have used a cross-over design (*Jajtner et al., 2015*; *Roberts et al., 2014*), given that the purpose of the current study was to examine the recovery effects of CWI and CAT in strength-untrained individuals, we separated participants into groups to avoid repeated bout effect or learning effects (*Doma et al., 2015*). Before commencing the study, each participant provided their written informed consent and did not report illness, disease and injury or medication that would contraindicate any protocols that were approved by the James Cook University Human Research Ethics Committee (HREC; Approval number H5565).
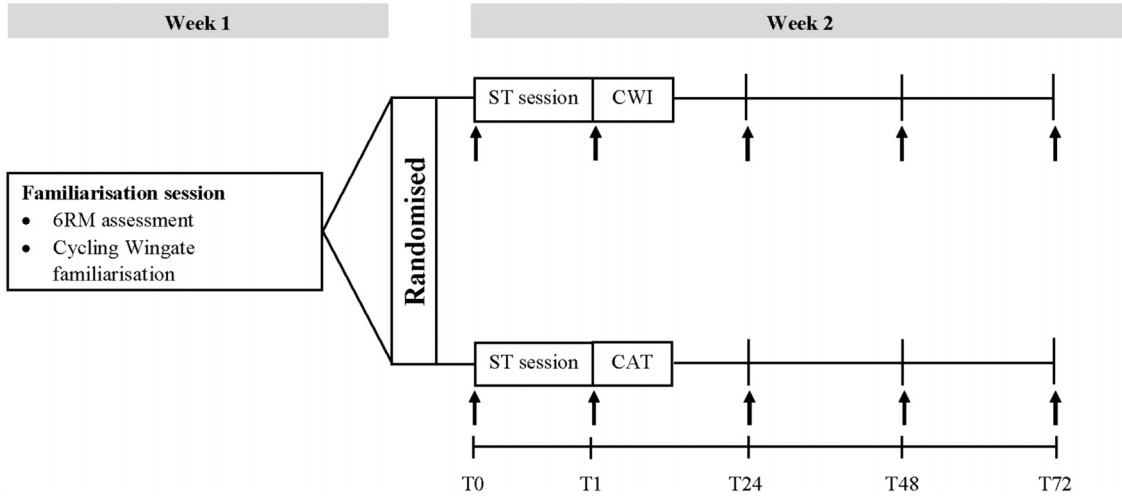

**Figure 1 Schematic of research design.** Schematic of the research design including cold water immersion (CWI) group and cold air therapy (CAT) group with indirect markers of muscle damage (i.e. creatine kinase, isometric knee flexor/extensor torque, muscle soreness and muscle fatigue) and anaerobic performance (i.e. cycling Wingate test) denoted in vertical arrows conducted prior to (T0), immediately post (T1), 24 (T24), 48 (T48) and 72 (T72) h post strength training (ST) session.

## Procedures

### Strength training session

The selection of exercises, intensity and duration of the strength training session was adapted from previous studies (*Doma & Deakin, 2013a*; *Doma & Deakin, 2014*) including leg-press, leg extensions and leg curls. These exercises were selected to replicate a typical lower body strength training session targeting maximal strength development (i.e. a low number of repetitions with high loads), typically recommended for endurance-trained subjects. In order to standardize the technique of each exercise, machine-based equipment was used. Specifically, participants underwent a warm-up set prior to their first working set of the strength training session by performing 10 repetitions of incline leg press using half the load of their 6RM. Participants then completed 5 sets of 6 repetitions of leg press at 6RM load and 3 sets of 6 repetitions of leg extensions and leg curls, both at 6RM load. A three minute recovery was provided between each set and each exercise.

### Maximal cycling performance

A 10-s Wingate test was performed on a bicycle ergometer (Velotron; Racermate, Seattle, WA, USA) using a standard protocol (*Minett et al., 2013*). Whilst a performance test of a longer duration, or over multiple sets, would have been desired, this cycling duration of a single set was selected to avoid additional fatigue across the post-strength training days (i.e. T24, T48 and T72). Furthermore, this duration has been used over a number of sets during high intensity intermittent cycling training to induce anaerobic capacity and improve sprint-ability (*Tabata et al., 1996*; *Wells et al., 2014*). Subsequently, attenuation of such performance measure, even with a single set, due to preceding strength

exercise-induced fatigue would suggest impairment of the quality of high intensity intermittent training sessions. Specifically, participants cycled at 50 W for 20 s, then cycled with the absence of any resistance for 5 s before finally cycling against a weighted brake (i.e., torque values of 0.087 and 0.084 for males and females, respectively) for 10 s. The participants were verbally encouraged during the protocol to ensure that cycling was executed at maximal effort.

### Indirect markers of muscle damage

Indirect markers of muscle damage were recorded via blood sampling (i.e., Creatine Kinase [CK]) and perceptions of muscle soreness and fatigue. Prior to analysing CK, calibration of the analyser (Reflotron Plus, Roche Diagnostics, Australia) was completed in accordance with the instructions of the manufacturer. Capillary blood samples were then collected using a finger prick method to analyse CK. Ratings of muscle soreness and fatigue were obtained following the completion of three repetitions of body weight squats. The muscle soreness and fatigue rating method used in this study was a standard 1–100 visual analogue scale adapted from a previous study (*Doma & Deakin, 2015*). Specifically, 1–100 indicated "not very sore at all" to "very sore," respectively, for the rating of muscle soreness and 1–100 indicated "not fatigued at all" to "very fatigued," respectively, for the rating of muscle fatigue.

### Muscle force generation capacity

Maximum Voluntary Isometric Contraction (MVC) testing was performed using a custom built isometric dynamometer chair (James Cook University, Cairns, Australia). The isometric dynamometer chair was set so as to position the right knee at an angle of 110°. A force transducer was placed superior to the right medial and lateral malleoli and calibrated by placing a known weight against the transducer. A high reliability (ICC = 0.76–0.95, mean 0.85) and minimal variability (CV = 4.1–5.9%, mean 5.1%) has been reported previously with this chair (*Doma & Deakin, 2014*). Participants were given three attempts at leg extensions and leg curls and were required to keep the maximal force plateaued for 6 s. A 90-s rest was allowed between attempts. The trial with the highest torque was then reported.

### Recovery protocol

Participants in the CWI group sat in an inflatable bath (White Gold Fitness, Bedford, UK) containing water at 14 °C for 15 min. To ensure that the lower extremity was fully submerged, the knees were extended and the water level was set to the iliac crest. To maintain the constant temperature of the water it was monitored using a thermometer and stirred at 5-min intervals with more ice added if necessary. The CAT group sat in the same bath without water in a custom-built climate chamber (James Cook University, Townsville, Australia) set at 14 °C and 60% humidity for 15 min. This humidity matched the ambient humidity that the CWI group were exposed to during the recovery intervention. Once positioned in the bath, blood pressure, heart rate and rating of thermal comfort were recorded at three minute intervals (i.e. five times) for safety monitoring

purposes. Thermal comfort was recorded using a 1–5 scale indicating "comfortable" to "extremely uncomfortable," respectively (*Minett et al., 2013*).

## Statistical analyses

The measure of central tendency and dispersion are reported as mean ± standard deviation. Given that the study incorporated male and female participants, inter-individual variability was controlled for by quantifying percentage differences from T1, T24, T48 and T72 against T0 for peak watts, mean watts and total work from the 10-s Wingate cycling test. The Shapiro-Wilk test revealed that all data analysed were normally distributed. Therefore, a two-way (time × group) repeated measures Analysis of Variance (ANOVA) was used to determine differences between each time point and between the CWI and CAT groups for the dependent variables with gender as a covariance. Bonferroni adjustments were performed for pairwise comparisons where significance was found. Significance was reached at an alpha level of $p \leq 0.05$. Effect Size (ES) was calculated to determine the magnitude of differences between the CWI and CAT groups for each time point (i.e. T0, T1, T24 and T48) using the formula:

$$ES = (CWI - CATmean)/standard\ deviation\ (Rhea, 2004).$$

Between-group ES for the indirect markers of muscle damage (i.e. CK, muscle soreness, muscle fatigue, KET and KFT) was calculated based on absolute measures. For the maximal cycling performance measures (i.e. peak watts, mean watts and total work), between-group ES was calculated based on the normalised data (i.e. percentage change across time). ES calculations were interpreted as trivial effect at <0.2, small effect at 0.2–0.49, moderate effect at 0.5–0.79 and a large effect at >0.8 (*Cohen, 1988*). All data were analysed using the Statistical Package for Social Sciences (SPSS, version 22, Chicago, Illinois).

## RESULTS

There were no significant differences between groups for age and strength levels ($p > 0.05$) indicating that the two groups were successfully matched for these parameters. Gender as a covariate showed no significant effects for all measures ($p > 0.05$) except for a time × gender interaction effect ($p < 0.05$) for CK, suggesting that gender did not influence the majority of the outcome measures for the two groups.

### Maximal cycling performance

A main effect for time was found for mean watts ($p < 0.01$) and total work ($p < 0.01$) but not peak watts ($p > 0.05$; Table 1). Similarly, these parameters showed no effect of time when examined per group (Table 2) with no time × group interaction effect ($p > 0.05$). However, effect size calculations between groups showed moderate to large differences between the groups (Fig. 2). At T24, a large difference was found between groups for peak watts, mean watts and total work. Furthermore, a moderate difference

**Table 1** Mean ± standard deviation for the main effect of time for the maximal cycling performance parameters measured prior to (T0), immediately post (T1), 24 (T24), 48 (T48) and 72 (T72) h post strength training session.

| | Mean power (W) | Peak power (W) | Total work (W) |
|---|---|---|---|
| T0 | 552.1 ± 94.25 | 741.15 ± 138.06 | 5079.27 ± 867.20 |
| T1 | 543.35 ± 98.14 | 693.75 ± 174.71 | 4999.34 ± 903.14 |
| T24 | 556.45 ± 96.61* | 750.65 ± 149.05 | 5119.46 ± 889.15* |
| T48 | 554.8 ± 98.55* | 754.70 ± 153.13 | 5104.32 ± 877.80 |
| T72 | 562.45 ± 95.68 | 765.25 ± 138.81 | 5171.77 ± 880.98* |

**Note:**
* Significantly greater than T1 ($p \leq 0.05$).

**Table 2** Mean ± standard deviation for the main effect of time for the maximal cycling performance parameters for the cold water immersion (CWI) and cold air therapy (CAT) group measured prior to (T0), immediately post (T1), 24 (T24), 48 (T48) and 72 (T72) h post strength training session.

| | Mean power (W) | | Peak power (W) | | Total work (W) | |
|---|---|---|---|---|---|---|
| | CWI | CAT | CWI | CAT | CWI | CAT |
| T0 | 541.9 ± 107.6 | 562.3 ± 143.0 | 727.7 ± 154.2 | 754.6 ± 212.0 | 4985.6 ± 989.7 | 5172.9 ± 1316.0 |
| T1 | 534.1 ± 117.2 | 552.6 ± 144.9 | 648.1 ± 247.8 | 739.4 ± 220.5 | 4913.4 ± 1078.3 | 5085.3 ± 1333.4 |
| T24 | 551.3 ± 113.4 | 561.6 ± 144.2 | 758.9 ± 185.1 | 742.4 ± 214.1 | 5768.8 ± 1043.7 | 5165.9 ± 1327.0 |
| T48 | 547.6 ± 113.1 | 562.0 ± 141.5 | 762.3 ± 170.2 | 747.1 ± 235.8 | 5037.5 ± 1041.4 | 5171.1 ± 1301.3 |
| T72 | 553.1 ± 113.7 | 571.8 ± 141.7 | 763.2 ± 150.2 | 767.3 ± 216.6 | 5087.6 ± 1044.6 | 5255.9 ± 1306.4 |

was found between groups at T48 and T72 for peak watts while a large difference was found at T48 and moderate difference at T72 for mean watts.

### Indirect markers of muscle damage

A main effect for time was found for CK, muscle soreness, muscle fatigue and knee extensor torque ($p < 0.01$) but not for knee flexor torque ($p > 0.05$; Table 3). Similarly, a main effect of time was found for CK, muscle soreness, muscle fatigue and knee extensor torque for the CWI ($p < 0.05$) and CAT ($p < 0.05$) groups but not for knee flexor torque ($p > 0.05$; Table 4). However, a time × group interaction effect was not found for any of these parameters ($p > 0.05$). Effect size calculations between groups only showed small to trivial differences following the treatment (Table 4). However, CK, muscle soreness and muscle fatigue were moderately larger for the CWI compared to the CAT group at T72.

### Thermal comfort

A main effect of time was found for thermal comfort ($p < 0.01$) with a time × group interaction effect ($p < 0.05$; Fig. 3). Specifically, time points on the 3rd min (2.43 ± 0.42) was significantly greater than the 6th (1.90 ± 0.31), 9th (1.45 ± 0.41) and 12th (1.35 ± 0.17) min. No main effect for group was observed ($p = 0.536$).

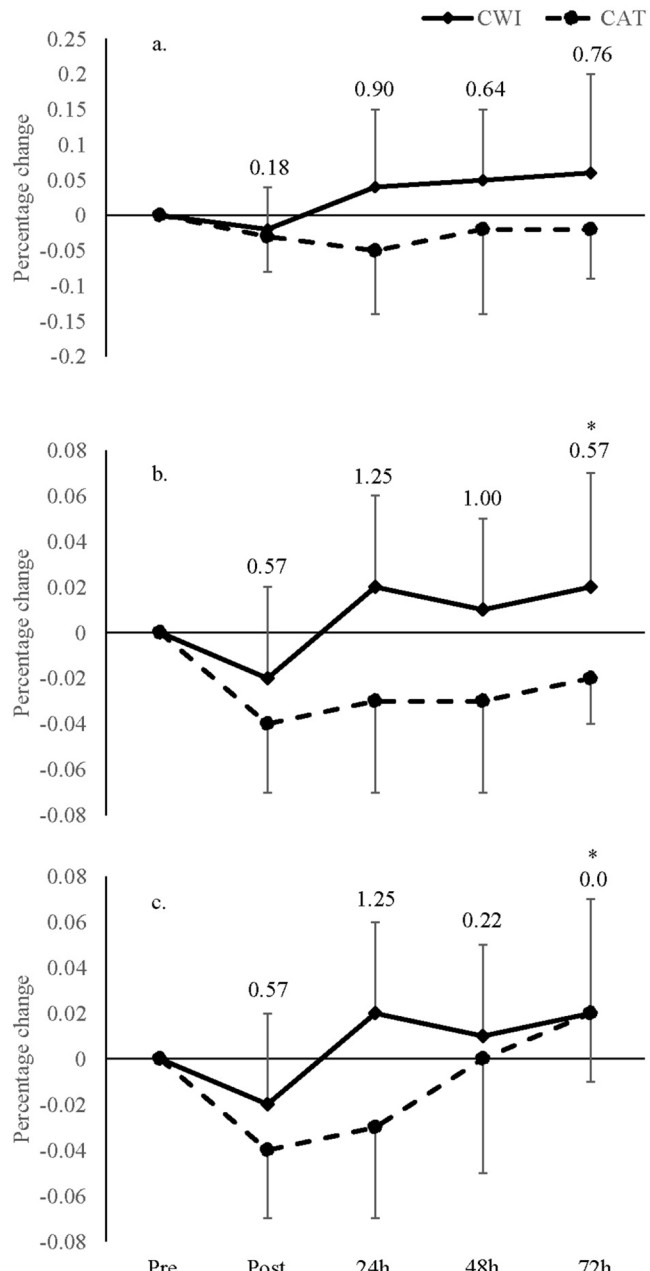

**Figure 2  Percentage change in cycling performance.** Percentage change in (A) peak watts, (B) mean watts and (C) total work for the cold water immersion (CWI) and cold air therapy (CAT) groups with effect size calculations between groups shown above bar graphs at immediate post (T1), 24 h (T24), 48 h. *Indicates significant increase from post (p ≤ 0.05).

## DISCUSSION

Whilst there were no statistical between-group differences, the ES analyses showed greater power output measures and total work done during cycling following the CWI compared to the CAT group with moderate to large differences at immediately post, 24, 48 and 72 h post strength training session. However, there were small differences between the CWI

**Table 3** Mean ± standard deviation for the main effect of time for creatine kinase (CK), muscle soreness, muscle fatigue, Knee Extensor Torque (KET) and Knee Flexor Torque (KFT) measured prior to (T0), immediately post (T1), 24 (T24), 48 (T48) and 72 (T72) h post strength training session.

|  | CK (U · L⁻¹) | Soreness | Fatigue | KET (N · m⁻¹) | KFT (N · m⁻¹) |
|---|---|---|---|---|---|
| T0 | 116.4 ± 45.42 | 4.25 ± 3.89 | 4.50 ± 4.55 | 199.32 ± 32.46 | 52.71 ± 17.68 |
| T1 | 252.83 ± 101.74 | 39.55 ± 15.14* | 47.55 ± 16.62* | 160.65 ± 26.83† | 50.51 ± 15.26 |
| T24 | 252.83 ± 101.42* | 47.55 ± 14.57* | 39.60 ± 16.88* | 178.05 ± 34.67 | 52.56 ± 15.63 |
| T48 | 173.25 ± 77.03* | 35.40 ± 14.60* | 27.9 ± 16.43* | 180.92 ± 36.59 | 52.85 ± 16.02 |
| T72 | 141.65 ± 50.04 | 17.50 ± 8.15§ | 12.90 ± 9.29§ | 183.10 ± 38.20 | 53.77 ± 17.51 |

Notes:
* Significantly greater than T0, T48 and T72.
§ Significantly greater than T0.
† Significantly lower than T0, T24, T48 and T72.

and CAT groups for indirect markers of muscle damage (i.e. CK, muscle soreness and muscle force generation capacity) at immediately post, 24, 48 and 72 h post strength training sessions. Whilst CWI only induced trivial effects on indirect markers of muscle damage, the moderate to large differences between the CWI and CAT groups for maximal cycling performance suggest that the combination of cold temperature and water submersion (i.e. CWI), than cold temperature alone (i.e. CAT), may alleviate fatigue and improve the quality of subsequent maximal sprint-based training sessions several days following strength training. In fact, a recent study showed that post-exercise cooling enhanced signalling pathways essential for mitochondrial biogenesis via the AMP-Activated Protein Kinase (AMPK) and p38 Mitogen-Activated Protein Kinase (MAPK) in human skeletal muscle (*Ihsan et al., 2015*). Accordingly, in addition to recovery of neuromuscular properties to improve sprint performance capacity which would remodel skeletal muscle to a more oxidative phenotype (*Cochran et al., 2014*), CWI appears to activate signalling pathways that may enhance training adaptation. However, chronic post-exercise cooling has also been shown to impair anabolic signalling that orchestrates strength training adaptation, which is also paramount for improving sprint-ability (*Saez de Villarreal et al., 2013*). Subsequently, further research is warranted to confirm whether CWI-induced recovery from strength training influences long-term sprint-based anaerobic development.

In the current study, power output did not change across time points with the only significant difference between T0 and T72. However, peak power output recovered better at T24, T48 and T72 for the CWI (4.0%, 5.0% and 6.0%, respectively) than the CAT (−5.0%, −2.0% and −2.0%, respectively) group and mean power recovered better at T24, T48 and T72 for the CWI (2.0%, 1.0% and 2.0%, respectively) than the CAT (−3.0%, −3.0% and 0.0%, respectively) group with moderate to large between-group differences. It is well known that a pressure gradient exists within the body, where blood and interstitial fluids flow from high to low pressure environments (*Rutkowski & Swartz, 2007*). Besides the beneficial effects of cooling, the purpose of CWI has long been to create a high pressure environment within the area of the body which has undergone muscle damage to increase removal of metabolites to areas of lower pressure

**Table 4** Mean ± standard deviation of creatine kinase (CK), muscle soreness, muscle fatigue, knee extensor torque (KET) and knee flexor torque (KFT) from pre (T0) to immediately post (T1), 24 h (T24), 48 h (T48) and 72 h (T72) post in the cold water immersion (CWI) and cold air therapy (CAT) groups with effect size calculations between groups for each time point.

| Variable | CWI | CAT | Effect size |
|---|---|---|---|
| CK (U · L$^{-1}$) | | | |
| T0 | 123.62 ± 75.05 | 109.18 ± 42.49 | 0.25 (small) |
| T24 | 259.39 ± 155.39* | 246.27 ± 113.47* | 0.10 (trivial) |
| T48 | 160.46 ± 106.85* | 186.03 ± 99.88* | 0.25 (small) |
| T72 | 157.88 ± 81.57 | 125.41 ± 48.7 | 0.50 (moderate) |
| Muscle soreness | | | |
| T0 | 3.8 ± 5.9 | 4.7 ± 6.2 | 0.15 (trivial) |
| T1 | 39.5 ± 20.3* | 39.6 ± 24.8* | 0.00 (trivial) |
| T24 | 47 ± 20.2* | 48.1 ± 23.2* | 0.05 (trivial) |
| T48 | 40 ± 20* | 30.8 ± 19.7* | 0.46 (small) |
| T72 | 20.9 ± 11.8* | 14.1 ± 9.8* | 0.63 (moderate) |
| Muscle fatigue | | | |
| T0 | 4.7 ± 7.4 | 4.3 ± 6.7 | 0.06 (trivial) |
| T1 | 54 ± 18.5* | 41.1 ± 26* | 0.58 (moderate) |
| T24 | 44 ± 21.3* | 35.2 ± 23.9* | 0.39 (small) |
| T48 | 32 ± 22.3* | 23.8 ± 21.1* | 0.38 (small) |
| T72 | 16.4 ± 14.2* | 9.4 ± 9.6 | 0.59 (moderate) |
| KET (N · m$^{-1}$) | | | |
| T0 | 203.7 ± 52.1 | 192.7 ± 49.6 | 0.15 (trivial) |
| T1 | 162.3 ± 39.6** | 156.8 ± 41.7** | 0.00 (trivial) |
| T24 | 180.8 ± 52.8 | 174.9 ± 53.7** | 0.05 (trivial) |
| T48 | 190.0 ± 58.6 | 180.4 ± 57.3** | 0.46 (small) |
| T72 | 190.8 ± 59.5 | 183.1 ± 57.6** | 0.63 (moderate) |
| KFT (N · m$^{-1}$) | | | |
| T0 | 51.5 ± 19.6 | 48.2 ± 19.1 | 0.06 (trivial) |
| T1 | 48.9 ± 20.3 | 45.7 ± 19.3 | 0.58 (moderate) |
| T24 | 50.1 ± 19.7 | 46.7 ± 19.4 | 0.39 (small) |
| T48 | 51.9 ± 20.7 | 47.9 ± 20.1 | 0.38 (small) |
| T72 | 53.4 ± 24.8 | 50.8 ± 25.0 | 0.59 (moderate) |

**Notes:**
* Significantly greater than T0 ($p < 0.05$).
** Significantly less than T0 ($p < 0.05$).

(*White & Wells, 2013*). Accordingly, the greater improvement in anaerobic capacity for the CWI compared to the CAT group in the current study suggests that hydrostatic pressure from water immersion appears to be a strong contributor to accelerating recovery than temperature alone. Indeed, including other experimental groups in different temperature and hydrostatic conditions would have further confirmed this and is considered a limitation in the study. Future research should compare recovery dynamics of CWI and CAT groups with groups in thermoneutral water and air conditions following lower body strength exercises.

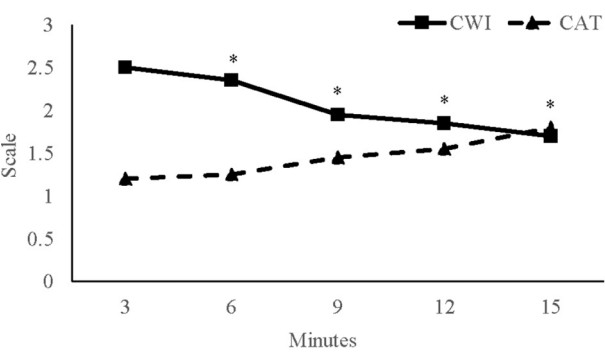

**Figure 3 Thermal comfort.** Thermal comfort for the cold water immersion (CWI) and cold air therapy (CAT) groups during the recovery interventions. *Indicates significant decrease from the 3rd min.

The improved anaerobic power in the CWI compared to the CAT group in the current study confirm those of *Vaile et al. (2008)* who reported improvement in vertical jump performance in the CWI group compared to a non-hydrotherapy group (i.e. passive recovery) at 48 and 72 h post strength exercises. However, there are methodological differences that should be elaborated upon. *Vaile et al. (2008)* determined power output from a single squat jump performance following a single exercise (i.e. leg-press) and incorporated a comparative group in thermoneutral conditions (i.e. warmer than CWI). Conversely, the current study examined maximal cycling performance following a strength training session consisting of multiple exercises (i.e. leg-press, leg extension and leg curl) with a non-hydrotherapy group in temperature conditions equivalent to the CWI group. Given that strength training volume (*Doma & Deakin, 2014*) and hydrostatic pressure (*Leeder et al., 2012*) have been shown to affect acute physiological responses, proper comparisons between the current study and that by *Vaile et al. (2008)* is at present difficult. Nonetheless, the current results extend those of *Vaile et al. (2008)* that CWI may improve lower body power output to a greater degree than non-hydrotherapy following lower body strength exercises.

Other than the current study, there has only been one study that has examined the effects of CWI on power output measures several days following a typical lower body strength training session consisting of multiple exercises (*Jajtner et al., 2015*) as far as we are aware. Interestingly, this study showed that the decrement in lower body power output were similar 24 and 48 h following strength training between the CWI ($-11.4\%$ and $-12.1\%$, respectively) and passive recovery ($-11.0\%$ and $-6.7\%$, respectively) group which conflict with the current results. The discrepancies in these findings could be attributed to a number of factors. Firstly, *Jajtner et al. (2015)* examined power output during squatting exercises at sub-maximal intensities with greater volume (i.e. three sets × ~10 repetitions of squatting exercises at 80% of 1RM). Whilst this method of assessment is important to demonstrate performance during a strength training session, sub-maximal squatting exercises may not be as responsive to fatigue as a single bout of maximal cycling performance assessment as that conducted in the current study. Secondly, *Jajtner et al. (2015)* incorporated strength-trained individuals with an average strength training history of 6.5 years. Conversely, the current study incorporated

endurance-trained individuals who had limited exposure to strength training. Subsequently, individuals in the current study may have experienced greater benefits from CWI, particularly on muscular performance, due to their lesser efficient recovery dynamics compared to individuals from the study by *Jajtner et al. (2015)*. This conjecture has been suggested by *Poppendieck et al. (2013)* where untrained individuals benefit from CWI to a greater degree due to potentially larger fatigue and soreness from reduced level of fitness. Furthermore, initial exposure to a typical strength training session has been shown to protect against muscle damage from the second session in strength-untrained men (*Doma et al., 2015*), known as the repeated bout effect. Accordingly, CWI may be more beneficial during the very initial stages of a concurrent training program for strength-untrained individuals where muscle damage is often excessive due to unaccustomed strength training exercises.

The CK measures between groups were comparable despite between-group differences shown for the maximal cycling performance measures. However, this discrepancy in trends between CK and dynamic performance measures following muscle-damaging protocols have actually been reported by others (*Chen & Hsieh, 2001*; *Chen et al., 2009*; *Chen, Nosaka & Tu, 2007*). Furthermore, previous studies have shown that blood levels of intramuscular proteins correlate poorly with changes in muscle function (*Clarkson et al., 1986*; *Newham, Jones & Clarkson, 1987*; *Newham, Jones & Edwards, 1983*). According to *Warren, Lowe & Armstrong (1999)* from their extensive narrative review on the appropriate measurement tools for muscle damage markers, they indicated that blood biomarkers do not accurately reflect conditions of the musculature, given that the level of intramuscular proteins in the systemic system may be representative of these compounds being released into the blood as well as being removed. Subsequently, *Warren, Lowe & Armstrong (1999)* suggested MVC as a better indicator of muscle damage than the measurement of intramuscular proteins alone. This further highlights the need to incorporate a range of indirect markers of muscle damage when monitoring the overarching condition of the musculature. Whilst every effort was made to account for potential gender differences, caution should be taken when interpreting our findings given that there was an interaction effect between CK and gender as a covariate. Furthermore, previous studies have reported gender differences in exercise-induced muscle damage and pain (*Dannecker et al., 2012*; *Fernandez-Gonzalo et al., 2014*).

Interestingly, when incorporating MVC in conjunction with CK in the current study, similar between-group trends were observed for these two indirect markers of muscle damage despite differences in maximal cycling performance measures. These findings are in line with others, where the trend in MVC was not reflective of changes in running performance measures following muscle-damaging exercises (*Chen et al., 2009*; *Chen, Nosaka & Tu, 2007*; *Doma & Deakin, 2014*) and also agree with others that have reported no differences between CWI and passive recovery for isometric strength measures (*Howatson, Goodall & van Someren, 2009*; *Sellwood et al., 2007*). Given that MVC in the current study was limited to examining contractile properties of the quadriceps and hamstrings, these measures may not have been a true reflection of the mechanics required for maximal cycling performance that requires multiple muscle groups

(e.g. quadriceps, hamstrings, gluteal muscles and gastrocnemi). Therefore, greater improvement in maximal cycling performance as a result of CWI may have occurred due to better recovery of muscle contractile properties required for cycling that were not examined in the current study. Further research is warranted to determine whether recovery dynamics other than knee extensor/flexor musculature would contribute to improvement in anaerobic power as a result of CWI.

*Leeder et al. (2012)* also reported CWI to induce greater improvement in power output measures but not on strength measures based on a meta-analysis of several CWI studies. The authors speculated that CWI may accelerate recovery of type 2 muscle fibres given that these fibre types are preferentially damaged as a result of strenuous exercises. However, *Vaile et al. (2008)* reported greater improvement in isometric squat performance and CK 48 and 72 h following strength training for the CWI compared to the passive recovery group. The discrepancies in findings between the current study and that by *Vaile et al. (2008)* could be attributed to the CWI protocol. The only discernible difference was the depth of CWI where *Vaile et al. (2008)* had participants immerse in CWI until the clavicle (i.e. entire trunk) compared to the current study where participants were immersed to the iliac crest. By exposing the torso to the CWI there may be an increased removal of metabolites and waste from the lower extremities to the thoracic cavity allowing for an increased rate of muscular regeneration in the affected muscles (*Versey, Halson & Dawson, 2013*).

## CONCLUSION

Overall, the current trends indicated that the application of CWI aided in the recovery of maximal cycling performance in strength-untrained but moderately endurance-trained individuals. These findings suggest that CWI may minimise the detrimental effects of lower body strength training-induced fatigue on the quality of subsequent high intensity intermittent training sessions, particularly during a concurrent training program. Given that strength training could induce sub-optimal training adaptations for modes of endurance exercise (*Dolezal & Potteiger, 1998*; *Psilander et al., 2015*; *Schumann et al., 2015*), improving anaerobic capacity via CWI may maximise training adaptations during concurrent training. This speculation could be confirmed by applying CWI between strength training and high intensity intermittent training sessions during a chronic training study (e.g. 10–20 weeks). However, caution should be taken given that speculation for the effects optimising training adaptation using CWI has been based solely on one bout of maximal cycling performance. Furthermore, future research should elucidate whether similar findings would be observed in anaerobically trained individuals and on other modes of exercise performance measures (e.g. running, rowing or swimming) to expand the practical application of cold-induced recovery methods. Finally, considering that typical CAT is set at extreme cold conditions with a much lower exposure window (i.e. under −100 °C for 2–3 min) possibly due to conductive/convective properties of the fluids, it would be interesting to compare intra-muscular temperature between CAT and CWI conditions similar to that in the current study.

### Funding

The authors received no funding for this work.

### Competing Interests

The authors declare that they have no competing interests.

### Author Contributions

- Kane J. Hayter conceived and designed the experiments, performed the experiments, analyzed the data, wrote the paper, prepared figures and/or tables.
- Kenji Doma conceived and designed the experiments, analyzed the data, wrote the paper, prepared figures and/or tables, reviewed drafts of the paper.
- Moritz Moritz Schumann conceived and designed the experiments, wrote the paper, reviewed drafts of the paper.
- Glen B. Deakin conceived and designed the experiments, analyzed the data, reviewed drafts of the paper.

### Human Ethics

The following information was supplied relating to ethical approvals (i.e., approving body and any reference numbers):

James Cook University Human Research Ethics Committee (HREC).

Approval number: H5565.

### Data Deposition

The raw data was supplied as a Supplemental Dataset File.

### Supplemental Information

Supplemental information for this article can be found online at http://dx.doi.org/10.7717/peerj.1841#supplemental-information.

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
