# Peer review of "The comparison of cold-water immersion and cold air therapy on maximal cycling performance and recovery markers following strength exercises"

_PeerJ, doi:10.7717/peerj.1841_

## Round 0.1 · original submission · Major Revisions

Introduction and discussion should be improved with some references suggested by the reviewers. In Addition to the points raised by the reviewers, please report some information in the introduction about cold air treatment (with relative citation) to strengthen the needs of comparison between CWI and CAT.

Both Reviewers have also highlighted some methodological issues (e.g. sample composition or familiarization). Please pay your attention on this part addressing all the comments before resubmitting the paper, as well as to all the other comments.

Moreover, please format the manuscript following the instruction of the Journal https://peerj.com/about/author-instructions/#standard-sections . In particular pay more attention to the abstracts’ sections.

·

Basic reporting

Pg. 17, line 266: “should be elaborated upon”.
Pg. 31: Notice that the legend (description) of the figure is not complete. Please check all figures.

Experimental design

Pg. 9, lines 104-107: More information is needed about the participants’ characteristics (e.g., runners or cyclists? Did they compete? What level?). What do you mean with “moderately endurance trained”? In addition, due to the reported sexual differences in response to muscle damaging exercises (e.g., Dannecker et al., 2012; Fernandez-Gonzalo et al., 2014), do you consider adequate (a priori) pooling men and women in your experimental design?
Dannecker, Erin A., et al. "Sex differences in exercise-induced muscle pain and muscle damage." The Journal of Pain 13.12 (2012): 1242-1249.
Fernandez-Gonzalo, Rodrigo, et al. "Muscle damage responses and adaptations to eccentric-overload resistance exercise in men and women." European journal of applied physiology 114.5 (2014): 1075-1084.
Pg. 9, lines 107-111: It is important to provide more details about the randomization procedures involved in the group allocation (e.g., did you use a computer-based random number generator? how was the matching of pairs of participants?).
Pg. 10, lines 126-127: Please provide more details about the choice of the strength training protocol (based on 6RM sets and lower body exercises).
Pg. 10, lines 131-143: You have provided a good rationale to use this protocol (i.e., avoid fatigue). However, due to the protocol characteristics, I am not convinced that possible impairments in the outcomes of this short-term anaerobic test would actually impair the “quality of high intensity intermittent training sessions” in endurance athletes. Please, address this issue.
Pg. 11, lines 155-164: Why exactly did you use MVC to quantify recovery in endurance trained participants? What would be the impact of reducing MVC on the quality of a subsequent endurance-based training session?

Validity of the findings

Pg. 14, lines 202-209: This paragraph needs clarification to improve reader’s understanding (please, revise the Statistical Analyses section accordingly). In tables 1 and 2 it is clear that the comparisons refer to the absolute values of mechanical performance obtained during the test. However, the figure 2 shows the percentage changes in the respective variables. What is not clear is whether the ES values refer to the between-group absolute values comparisons or to the percentage changes. Considering the former, an ES comparing the pre- values is missing. Contrariwise, you need to state in the Statistical Analyses that the ES was calculated from the comparisons involving the percentage changes (with which I agree).
Pg. 15, line 221: Again, I am a bit confused. Here, it appears that you are comparing the absolute values (and not the percentage changes) using the ES. Accordingly, the pre (T0) value was included in the analyses.
Pg. 15, lines 234-237: These findings need to be confirmed by the clarification of the statistical analyses mentioned above.
Pg. 16, lines 243-244: Suggest more cautious while stating something about endurance performance, since you have used a very short-term test to assess performance (i.e., it does not even involve a significant cardiorespiratory response).
About the Discussion section in general - I would be happy to see the opinion of the authors about the markers used to assess muscle damage. You have found differences (using ES) between CWI and CAT (which need to be clarified by the statistical analyses used) in the mechanical performance in an anaerobic test. However, the other markers of “muscle damage” did not present a time vs. group interaction. In this case, muscle damage was or was not observable after the strength training session?
Pg. 20, line 329: At this point, you use the term “fatigue” instead of “damage”, but the objective of the study was to test the effects of CWI or CAT on muscle damage markers.

Additional comments

A recent study has shown a substantial “placebo” effect related to the use of CWI (Broatch et al., 2014). Considering the results of this study, what is the potential bias of participants’ beliefs regarding the effectiveness of CWI or CAT?
Broatch, James R., Aaron Petersen, and David J. Bishop. "Postexercise cold-water immersion benefits are not greater than the placebo effect." Med. Sci. Sports Exerc 46.11 (2014): 2139-2147.
Studies using CAT make use of significantly lower temperatures in the chambers. This is probably related to the conductive/convective properties of the fluids (air vs. water). Don’t you think that part of your results favoring CWI in relation to CAT can be explained by this fact (differences not only in the presence or not of hydrostatic pressure but also differences in the cooling of the muscle tissues)?

Reviewer 2 ·

Basic reporting

No comments

Experimental design

Authors should report also heart rate and blood pressure values (that even thought not a dependent variables, nevertheless could be of interest for readers) and report the devices used for the measurements

Validity of the findings

Results sounds convincing but I strongly suggest to expand the part regarding the not significant results of muscle damage blood marker. This is a very interesting point that worth deeper discussion: why there are no difference in muscle damage markers but there are in performance, how muscle damage can influence performance?

Additional comments

Abstract line 31 please change exercises with "training session"
Methods, first paragraph, line 91. Are you sure that 4 days are enough to recovery from the familiarization session? I am not so sure... Please discuss this point
I strongly suggest to ad a discussion about the relevance of authors' results compared to two recent papers "Ihsan 2015 Regular postexercise cooling enhances mitochondrial biogenesis through AMPK and p38 MAPK in human skeletal muscle" and "Roberts 2015 Post-exercise cold water immersion attenuates acute anabolic signalling and long-term adaptations in muscle to strength training". CWI seems to increase mitochondrial biogenesis but at the same time reduce anabolic response. These data should be discussed and authors' comments could be very interesting for the readers

---

## Round 0.2 · accepted · Accept

I'm glad to inform you that you have improved the manuscript and replied to all the comments raised by the reviewers.

·

Basic reporting

No comments

Experimental design

No Comments

Validity of the findings

No comments

Additional comments

The authors have addressed all the comments/suggestions adequately. I am happy with the version submitted after revision. Findings are valuable to other research groups investigating recovery methods aiming at enhancing training adaptations in endurance athletes.

Reviewer 2 ·

Basic reporting

The article is suitable for publication

Experimental design

The article is suitable for publication

Validity of the findings

The article is suitable for publication

Additional comments

Authors answered to all my concerns